# Impact of Rapid Response Teams on Pediatric Care: An Interrupted Time Series Analysis of Unplanned PICU Admissions and Cardiac Arrests

**DOI:** 10.3390/healthcare12050518

**Published:** 2024-02-21

**Authors:** Samah Al-Harbi

**Affiliations:** 1Department of Pediatrics, Faculty of Medicine, King Abdulaziz University, Jeddah 21589, Saudi Arabia; smhalharbi@kau.edu.sa; 2Department of Pediatrics, King Abdulaziz University Hospital, Jeddah 22252, Saudi Arabia

**Keywords:** pediatrics, patient care team, intensive care units, time series studies, cardiac arrest, before-and-after study

## Abstract

Pediatric rapid response teams (RRTs) are expected to significantly lower pediatric mortality in healthcare settings. This study evaluates RRTs’ effectiveness in decreasing cardiac arrests and unexpected Pediatric Intensive Care Unit (PICU) admissions. A quasi-experimental study (2014–2017) at King Abdulaziz University Hospital, Jeddah, Saudi Arabia, involved 3261 pediatric inpatients, split into pre-intervention (1604) and post-intervention (1657) groups. Baseline pediatric warning scores and monthly data on admissions, transfers, arrests, and mortality were analyzed pre- and post-intervention. Statistical methods including bootstrapping, segmented regression, and a Zero-Inflation Poisson model were employed to ensure a comprehensive evaluation of the intervention’s impact. RRT was activated 471 times, primarily for respiratory distress (29.30%), sepsis (22.30%), clinical anxiety (13.80%), and hematological abnormalities (6.7%). Family concerns triggered 0.1% of activations. Post-RRT implementation, unplanned PICU admissions significantly reduced (RR = 0.552, 95% CI 0.485–0.628, *p* < 0.0001), and non-ICU cardiac arrests were eliminated (RR = 0). Patient care improvement was notable, with a −9.61 coefficient for PICU admissions (95% CI: −12.65 to −6.57, *p* < 0.001) and a −1.641 coefficient for non-ICU cardiac arrests (95% CI: −2.22 to −1.06, *p* < 0.001). Sensitivity analysis showed mixed results for PICU admissions, while zero-inflation Poisson analysis confirmed a reduction in non-ICU arrests. The deployment of pediatric RRTs is associated with fewer unexpected PICU admissions and non-ICU cardiopulmonary arrests, indicating improved PICU management. Further research using robust scientific methods is necessary to conclusively determine RRTs’ clinical benefits.

## 1. Introduction

### Background

Indeed, due to recent advancements in pediatric care and medical technology, the proportion of children who now survive once-fatal disorders that were once deadly has grown dramatically, changing the epidemiology of pediatric [1,2] ailments that affect between 0.4% and 0.7% of all children and account for 42% of pediatric inpatients and up to 53% of PICU admissions at any given time [3]. Similar situations can be found in wealthier countries [4]. As a result, hospitalized pediatric patients frequently experience medical emergencies [5,6]. These include unplanned ICU admission, cardiac arrest while hospitalized, and death. To begin, cardiorespiratory arrest in children is a rare but tragic complication of severe sickness in hospitals [7] P. Suominen (2000), for example, observed 0.7% cardiopulmonary arrest among hospitalized children [8], Song et al., (2011) 3% [9], J. Zeng (2013) 0.18% [10], A. Latif (2014) 3% [11], and V. Rathore (2016) 0.7–3% [12]. Additionally, the Pediatric Data Quality System Collaborative Measure Workgroup selected the unplanned PICU admission rate as one of the quality metrics to be used in a systematic evaluation to identify quality improvement activities [3,13]. It raises the expense of living, morbidity, and mortality. Unfortunately, this is a common occurrence. For example, Kathryn O. Mansell (2018) reported 105/450 (23.3%) unscheduled PICU transfers [14], and da Silva (2013) 28/116 (24.1%) [15]. It could range from 30% to 57% [16]. Ward patients admitted to the PICU had a higher death rate (odds ratio 1.65, 95% confidence interval 1.08–2.51) and lasted nearly four days longer than emergency department patients [17]. According to El Halal, Michel Georges Dos Santos (2012), and other contributors, mortality doubles [18]. Similar findings were found in other publications (2.78% versus 1.95%; *p*-value 0.001), the hazard ratio was 1.71 (95% CI: 1.5 to 1.9), and the PICU stay was longer (4.9 versus 3.6 days; *p* = 0.001) [19].

To Err Is Human contends that the problem in health care is not bad individuals; rather, decent people operate in unsafe systems that must be improved. Since the publication of “To Err Is Human: Creating a Healthier System” [20], two new papers recognize improvements while arguing that we still have a long way to go to ensure that care is as safe as it should be for all patients [21]. Undetected clinical decline in hospitalized children frequently leads to preventable adverse outcomes and constitutes a major source of avoidable harm [22]. As a result, healthcare institutions have pledged to invest in patient safety programs [11,23,24,25]. One such initiative was the establishment of a pediatric rapid response team, a critical domain of emergency patient management in the accreditation process of the healthcare business [26].

Pediatric rapid response teams (RRTs), especially those led by clinicians with experience in critical care, have been shown to significantly improve the quality of patient care in pediatric settings. This enhancement is primarily achieved through the timely identification and management of patient deterioration, thereby mitigating the risk of adverse outcomes. RRTs, as a proactive measure, have been identified as a pivotal strategy in diminishing mortality rates within hospital settings, showcasing the indispensable benefits of RRTs in the healthcare landscape of healthcare [27]. Moreover, the introduction of Rapid Response Systems (RRSs) has been associated with a significant elongation in the interval between cardiac arrests, serving as a testament to the system’s efficacy in pre-empting severe adverse events [28,29]. This important development makes it even more clear how important RRTs are for improving patient safety and outcomes by making early interventions easier and stopping conditions from getting worse. Prompt activation of pediatric RRTs is now recognized as a cardinal standard in the management of hospitalized patients, playing a crucial role in the attenuation of cardiac arrests and the reduction of unforeseen ICU admissions. This underscores the fundamental importance of RRSs in fostering safer hospital environments. In contrast, delays in the activation of RRSs have been linked to negative repercussions on the patient’s clinical trajectory, highlighting the imperative for swift and decisive action to achieve optimal patient outcomes [29].

Nursing staff can identify children at imminent danger of clinical decline by carefully evaluating vital signs, patient symptoms, and other clinical markers using Pediatric Early Warning Scores (PEWS) [30,31,32]. By utilizing a PEWS framework—which includes a complex escalation algorithm—medical staff are notified when high-risk patients require urgent bedside assessments. There is much empirical evidence that shows how pediatric rapid response teams can reduce in-hospital mortality, cardiac arrests, hospital and intensive care unit stays, and related financial burdens by including PEWS in their intervention strategies [33,34,35]. The tertiary academic medical institution King Abdulaziz University Hospital in Jeddah, Saudi Arabia, has taken the lead in a quality improvement project, utilizing a pediatric quick response team, with the goal of improving patient safety and care standards. It is worth mentioning that this study is the first of its kind to employ the interrupted time series model to thoroughly examine the impact of the pediatric rapid response team on the reduction of non-ICU cardiorespiratory arrests, unplanned admissions to the pediatric intensive care unit, and mortality. This study aims to systematically examine the efficacy of Rapid Response Teams (RRTs) in diminishing the rates of unplanned Pediatric Intensive Care Unit (PICU) admissions and cardiopulmonary arrests. By meticulously documenting and quantifying the benefits of RRTs on pediatric patient outcomes, it aspires to establish a foundational benchmark for subsequent research and clinical practices.

## 2. Methods

### 2.1. Study Design and Setting

The research took place at King Abdulaziz University Hospital in Jeddah, Saudi Arabia. The pediatric department at this facility has fifty beds and handles about a thousand pediatric admissions per year. We employed a before-and-after technique, rather than random assignment to the intervention and control groups, to determine the efficacy of pediatric rapid response teams (RRTs). To put it simply, this is a quasi-experimental layout. There were 3261 pediatric inpatients included in the study, which ran from 1 January 2014 to 30 July 2017. Two separate periods made up the cohort: the first, beginning on 1 January 2014 and ending on 30 August 2015, included 1604 patients; the second, beginning on 1 September 2015 and ending on 30 July 2017, had 1677 patients. Because the intervention was implemented in a real-life clinical context, this study can be considered an experiment. Randomized controlled trials are not always feasible due to ethical, logistical, or budgetary concerns. We were able to demonstrate how RRTs affected the frequency of PICU admissions and cardiac arrests using this design, which allowed us to examine their use in a real-world context while still accounting for the limitations of non-randomized trials.

### 2.2. Participants and Inclusion Criteria

This investigation encompassed all pediatric inpatients within the age range of one month to fourteen years admitted to the general pediatric ward, irrespective of the admission cause, with the explicit exclusion of neonates and individuals designated as ‘no code’. This inclusion strategy was meticulously designed to yield a comprehensive examination of the pediatric cohort within the specified ward, aligning with the rigorously defined scope and exclusion parameters of our study. To ensure methodological rigor, the purview of our research was strictly limited to the general pediatric ward. Consequently, ancillary sectors such as the Pediatric Intensive Care Unit, the emergency department, the post-operative unit, and the operation theatre were deliberately excluded from the scope of our analysis.

### 2.3. Ethical Considerations

The conduct of this study was rigorously aligned with ethical standards and received formal approval from the Institutional Review Board of the King Abdulaziz University, Jeddah, as documented under reference number 79-23. Recognizing the quasi-experimental design and retrospective nature of the research, all ethical recommendations pertinent to such studies were meticulously adhered to. This included a thorough review by the ethics committee, ensuring that the study met the required ethical guidelines for quasi-experimental research, particularly in relation to data confidentiality, patient safety, and the integrity of the research process. Although the retrospective design of the study negated the need for informed consent, all patient information was handled with the utmost confidentiality and in compliance with ethical standards for medical research, safeguarding the rights and well-being of the participants involved.

### 2.4. Composition of the Pediatric Rapid Response Team (RRT)

At our institution, the pediatric rapid response team (RRT) functions under the aegis of a dedicated lead physician and consists of a multidisciplinary cohort. The team structure includes a Pediatric Intensive Care Unit (PICU) attending physician, a specialty physician, and, depending on the shift, a daytime fellow/resident or an overnight PICU fellow/resident, all under the supervision of an attending physician. The team’s capability is further bolstered by a critical care nurse and a respiratory therapist. A distinguishing feature of our RRT is the exclusive allocation of nurses to the team, ensuring focused and uninterrupted rapid response care. While the primary commitment of the physicians and other team members is to the RRT, they also extend their support to the PICU during nighttime hours as needed. To address and minimize potential variability in patient assessment and care provided by different team members, each member of the RRT has undergone specialized training designed to standardize and optimize team performance. This rigorous training ensures consistency in the evaluation process and the execution of the care plan across different shifts and scenarios. Consequently, this structured approach mitigates the risk of variation in care delivery, ensuring that all pediatric patients receive the highest standard of rapid response intervention, irrespective of the time of day or the specific team members on duty.

### 2.5. Activation and Scoring Mechanism

With the help of a specialized pager system, the Rapid Response Team (RRT) may be quickly summoned to assist any patient on the ward who need immediate attention. The initiative’s evaluation methodology is founded on the ‘Modified Pediatric Early Warning Score’ and the ‘COAT score’ (CHILDREN’S OBSERVATION AND ALERT TOOL), which are adaptations of the PEWS scoring matrix from the NHS Institute for Innovation and Improvement. It should be emphasized that the COAT tool is meant to supplement, not replace, the priceless knowledge that can be obtained from experienced clinical judgment and thorough patient evaluation. Within a COAT score range of 0–26, this evaluation technique thoroughly examines nine important clinical factors. An intuitive way to understand these scores is by looking at the color codes that correspond to different age groups; higher COAT scores usually mean that kids’ health is getting worse. According to the COAT score gradient, a well-organized plan of action should be put into place when the score is 3 or above; scores between 4 and 6 should be considered for RRT activation; and scores of 7 or higher should prompt the rapid reaction team to be deployed immediately. Research confirming these instruments highlights their effectiveness. Extensive research has shown that the Modified Pediatric Early Warning Score (PEWS) can effectively identify pediatric patients who are at risk of experiencing clinical worsening, making it an essential tool for pediatric patient surveillance. The sensitivity and specificity for a PEWS score of 3 were found to be high, according to Rosman et al., (2019) (96.2%). It had excellent diagnostic performance, with a high sensitivity (82% at an 8-point scale) and specificity (93% at a 9-point scale), as reported by Parshuram et al. [36]. The area under the receiver operating characteristic curve was 0.91. The strong discriminant ability of the PEWS score for hospital admission was highlighted by Breslin et al., (2014), who found that a 1-point rise in the score greatly increased the probability of both acute and intensive care admissions, particularly for patients with respiratory problems [37]. Fujikoshi et al., (2015) showed that the score was useful for predicting the requirement for unplanned admission to the pediatric intensive care unit [38], further validating it. As Vredebregt et al., (2019) pointed out, it could rule out critical illness in children visiting emergency rooms, and its reliability was validated with a sensitivity of 85% and a specificity of 80%. It is worth noting that there was an encouraging trend when the Manchester Triage System was combined with the Modified PEWS, although this combination did not substantially surpass the performance of the Triage System alone [39]. All things considered, these studies prove that the Modified Pediatric Early Warning Score is an essential part of the Rapid Response Team’s structure, and that it is invaluable for providing proactive and accurate pediatric care (Table 1).

### 2.6. Data Acquisition

The data-collecting process for this study was well-organized to guarantee that all pertinent parameters were captured accurately both before and after the introduction of Rapid Response Teams (RRTs) to pediatric care. A retrospective evaluation of the time before the intervention and a prospective evaluation of the time after the intervention were the two separate parts of the data gathering process. The main data repository was Electronic Medical Records (EMRs), which ensured a consistent and dependable framework for data collecting. Extraction of historical data from EMRs was carried out with great care throughout the pre-intervention phase. Important benchmark scores from the Pediatric Early Warning Signs (PEWS) assessment, which quantify the severity of the patient’s condition before the intervention, were among the data points included in this set. The number of pediatric ward admissions, patient demographics (age and gender), PICU transfers, cardiorespiratory arrests, admissions to the PICU, and mortality rates were among the many metrics that were reviewed retrospectively. Consistency and comparability of data across both timelines were ensured by adhering to the same data collection protocol in the prospective period, post-intervention. This phase began with the deployment of RRTs and continued with the real-time collection of data to evaluate the immediate and changing effects of this intervention on pediatric patient outcomes. The data collection method was handed to a multi-disciplinary team that included pediatric care specialists, nursing staff, and data analysts. To ensure the security and privacy of patient information, the team adhered to rigorous processes. We used a stringent validation procedure to ensure the data was reliable. To do this, we compared the acquired data with medical records and ward reports. The staff involved in data gathering were swiftly consulted during a consultative review to correct any anomalies.

### 2.7. Outcome Measures

The primary outcome indicators were the number of pediatric ward patients shifted to the PICU and non-ICU cardiorespiratory arrests per month. Outside of the ICU, a code was defined as any general ward patient who required tracheal intubation, chest compressions, or both respiratory and cardiopulmonary arrest. The secondary outcome for patients transported to the PICU after the quick response system is activated is mortality.

### 2.8. Statistical Analysis

To simplify the baseline characteristics of the dataset, the R package ‘table one’ summarized all numerical and categorical variables in detail. Counts and percentages were used to represent category variables. Due to the non-normal distribution of some of the continuous variables, we opted to summarize them using median and interquartile ranges rather than means and standard deviations. Our comparison tests assumed that the data was normally distributed. We used Fisher’s exact test as necessary, but the chi-square test was our go-to for categorical variables. Because of their similarity in comparing means across groups, we used *t*-tests or two-group ANOVAs where the data followed a normal distribution. Nevertheless, non-parametric alternatives were utilized when dealing with continuous variables that did not conform to the normalcy assumptions. When comparing more than two groups, we used the Kruskal–Wallis test. When comparing two groups, we used the Wilcoxon rank-sum test, which is comparable to the Mann–Whitney U test. 

The statistical study employed an interrupted time series (ITS) technique to ascertain the intervention’s impact on the outcome variable. Two metrics were examined as part of this process: the level change, which displays the result’s change immediately after the intervention, and the slope change, which shows the pattern of the outcome’s change. The ITS investigation meticulously counted coefficient estimations, taking into consideration changes in level and slope both before and after the intervention. Calculating p-values and confidence ranges for each coefficient allowed us to verify if the results were robust. These details illuminated the veracity and statistical importance of the observed alterations. We computed the coefficient for both the pre- and post-intervention times, together with standard errors, *t*-values, and 95% confidence intervals, to evaluate the magnitude and significance of the intervention’s effect. Using a *p*-value threshold of less than 0.05, we examined whether the changes we saw were statistically significant. 

Test for Sensitivity: By integrating state-of-the-art statistical methods with an interrupted time series (ITS) framework, we conducted a thorough sensitivity analysis. The initial purpose of using bootstrapping was to assess the stability of the ITS model estimates for unscheduled PICU admissions. We were able to achieve this by resampling and recalculating the model’s coefficients multiple times to see how the effect size and direction changed over time. As an additional measure, the outcomes of the bootstrapping procedure were verified using segmented regression analysis. The sensitivity test also used a Zero-Inflation Poisson (ZIP) model to look at the data after the intervention, which showed that there were too many zeros in the non-ICU cardiopulmonary arrests.

## 3. Results

This investigation encompasses pediatric inpatients who were admitted for at least one day to a medical ward from 1 January 2014 to 30 July 2017. The cohort consisted of 3281 pediatric patients, divided into pre-intervention (1604) and post-intervention (1677) groups. The pre-intervention phase spanned from 1 January 2014 to 30 August 2015, while the post-intervention phase covered 1 September 2015 to 30 July 2017. Analysis revealed that the median Pediatric Early Warning Score (PEWS) for both cohorts was 6 (Interquartile Range [IQR] 5–7), yielding a *p*-value of 0.217. Gender distribution was similar across the groups, with females comprising 45.2% (725) and 46.5% (780) in the pre- and post-intervention phases, respectively, with a *p*-value of 0.45. The median age shifted from 6 months (IQR 2–12) pre-intervention to 7 months (IQR 3–24) post-intervention, with a *p*-value of 0.31.

Significantly, the median monthly admission rate from pediatric wards to the Pediatric Intensive Care Unit (PICU) decreased from 21.5 [20.25–25.75] to 13 [10,11,12,13,14,15,16,17,18], with a *p*-value of 0.009. Concurrently, occurrences of non-ICU cardiopulmonary arrests reduced from a median of 2 [0.25–3] to zero, as evidenced by a *p*-value of 0.001 in bivariate analysis (Table 2). Throughout the study, 471 pediatric rapid response teams (PRRTs) activations were recorded, with respiratory distress (138/471, 29.30%), sepsis (105/471, 22.30%), and physician concern (65/471, 13.80%) being the predominant triggers. 

Comparative analysis of PICU admissions pre- and post-intervention revealed a decrease from 487 out of 1604 patients (30.5%) to 281 out of 1677 patients (16.8%), resulting in an unadjusted relative risk of 0.55 (95% CI: 0.48–0.63, *p*-value < 0.0001) (Table 3). Prior to the intervention, there were 24 cases of cardiorespiratory arrest among 1604 ward patients, whereas no events were reported in the post-intervention cohort of 1674, indicating a complete mitigation of such incidents post-intervention. (Table 4). The PRRT’s activation also significantly reduced the unadjusted relative risk of mortality to 0.08 (95% CI: 0.03–0.26, *p*-value < 0.0001).

Notably, the proportion of PICU admissions originating from wards shifted from 62.5% (487/779) pre-intervention to 33.6% (281/837) post-intervention. Additionally, PICU-observed crude mortality rates declined from 7.7% to 3.0%. An interrupted time series (ITS) analysis underscored a significant association between the PRRT intervention and a reduction in unplanned PICU admissions, as indicated by a coefficient of −9.61 (95% CI: −12.65 to −6.57, *p*-value = 0.001). This translates to a reduction of 9.61 unplanned PICU admissions per unit time (Figure 1 and Table 5). Similarly, the intervention correlated with a significant decrease in non-ICU cardiac arrests, as reflected by a coefficient of −1.641 (95% CI: −2.22 to −1.06, *p*-value = 0.001), equating to 1.64 fewer incidents per unit time (Figure 2 and Table 5). These outcomes robustly affirm the effectiveness of the PRRT intervention in enhancing key pediatric healthcare metrics.

### Sensitivity Test

Pre- and post-intervention bootstrapped coefficient distribution for unplanned Pediatric Intensive Care Unit (PICU) admissions. This graph depicts the range of coefficients discovered using bootstrapping technique (N = 1000 resamples) to determine how a healthcare intervention affects persons who end up in the PICU unexpectedly. The original effect is depicted by a red dashed line with a mean difference of −9.61 in unplanned PICU admissions, demonstrating a decrease post-intervention. The green dashed lines represent the 95% confidence interval, which ranges from −3.65 to 4.04 and represents the variability in the estimated intervention impact (Figure 3). This interval’s broad range includes both negative and positive values, indicating significant variation across resampled datasets. This variation emphasizes how difficult it is to determine the precise impact of the intervention because it implies that, while there is an association with fewer PICU admissions, the strength and stability of this effect are variable and may depend on factors such as data noise, sample size, and the specifics of the intervention. Segmented Regression Analysis revealed a similar finding (Figure 3). The impact of the pediatric rapid response team intervention is revealed by the Zero-Inflated Poisson (ZIP) model analysis for sensitivity testing for non-ICU cardiac arrest data. The intervention had a substantial beneficial effect on the zero-inflation component, which estimates the chance of zero occurrences (estimate = 5.081, *p* < 0.001). This significant finding suggests that the intervention significantly increased the probability of having no post-intervention non-ICU cardiac arrests, thereby lowering their overall occurrence.

## 4. Discussion

This single-center, before-and-after, quasi-experimental cohort study at King Abdulaziz University Hospital in Jeddah, Saudi Arabia, reported the establishment of a PICU physician-led fast response team. This study aimed to determine the causal link between the establishment of a pediatric rapid response team and primary outcomes such as unexpected PICU admission and non-ICU cardiorespiratory arrest. In addition, it aimed to investigate if the death rate of pediatric ward patients transferred to the PICU fell when the pediatric rapid response system was implemented. In our study, the median PWES score for both groups are 6 (IQR 5–6.75) and 6 (IQR 5–7), with a *p*-value of 0.217, respectively. Values for gender were female 725 (45.2%) versus 780 (46.5%) with *p* value 0.45. The value for pre-intervention median age was 6 (IQR 2–12), and for post-intervention 7 (IQR 3–24) months, *p* value 0.31 (Table 2). Hence, the disease severity score, age, and gender are matched in both groups. During our study period, 471 pediatric rapid response systems were activated throughout a 21-month period, corresponding to 281 rapid response team activations per 1000 pediatric admissions. In our analysis, we found that respiratory distress was the leading cause of pediatric rapid response team activation (138/471, 29.30%), whereas previously published data ranged from 36.3% to 55.9% [40,41,42,43]. Furthermore, sepsis (105/471, 22.30%) is the second most prevalent cause, with a higher incidence reported by White K, Scott IA, and others (2016) at 29.5% [44], a lower rate than ours at 16.1% [40], and a higher incidence was reported by Lockwood JM, Ziniel SI, and Bonafide CP (2021) at 44% [45]. The pediatric rapid response team is an excellent tool for initiating early goal-directed therapy for sepsis and lowering sepsis-related mortality [46].

Crossing the Quality Chasm, a 2001 publication, encourages healthcare professionals and policymakers to implement new system methodologies and ideas to accomplish change. One of these ideas was for hospitals to form a rapid response team that family members could call if they were worried about their children [47,48]. In our research, family participation in the pediatric rapid response system was extremely poor (3/471, 0.64%). They are unmotivated to share responsibility for their sick offspring. Family involvement is currently critical to patient safety, especially in pediatric rapid response systems [49]. We know that parents are more likely than experienced physicians and nurses to notice significant changes in their child’s temperament and behavior. Transfer to the ICU is less common when family members activate the fast response team (25% vs. 59%, *p*-value 0.001) [50] and mortality is reduced [50]. In a survey in the USA, 69% of families wanted to be involved in pediatric rapid response team activation [51]. As a result, the National Patient Safety Organization and the Josie King Foundation have joined forces to support the family-led activation of rapid response team [52]. In contrast, Albutt et al., (2016) cited that currently there is insufficient evidence demonstrating that a family-activated escalation service is the most effective way to prevent patient deterioration [53]. Other noteworthy causes of emergency system activation besides respiratory and sepsis were physician concern (65/471, 13.80%), hematology (32/471, 6.80%), cardiovascular (31/471, 6.58%), neurology (23/471, 4.88%), and post-operative complications (9/471, 1.91%).

To the best of our knowledge, this is the first study that examines the causal inference concerning the pediatric rapid response team’s influence on the reduction of unplanned PICU admissions and pediatric non-ICU cardiorespiratory arrest using the interrupted time series model [54]. The findings of the “100,000 Lives Campaign” prompted the Institute for Healthcare Improvement to propose using rapid response teams to improve patient safety outcomes [55]. Furthermore, the Joint Commission mandated that United States hospitals should establish rapid response system (RRSs) as part of the 2008 National Patient Safety Goals [56]. There was some disagreement: Winters et al. claimed that there is weak evidence that rapid response systems are associated with a reduction in hospital mortality and cardiac arrest rates based on a systemic review, but they acknowledged that the presence of heterogeneity limited their ability to make such an assumption [57]. Similarly, Hillman and Ken (2005) identified no statistically significant difference in cardiac arrest rates (1.64 vs. 1.31, *p* = 0.736) or unplanned ICU hospitalizations (4.68 vs. 4.19, *p* = 0.599) [58].

Unscheduled admissions to pediatric critical care units represent a significant challenge within the healthcare sector. Data from the Pediatric Intensive Care Audit Network reveal that in the United Kingdom, sudden clinical deteriorations were responsible for unexpected critical care admissions in upwards of 85% of instances in 2017 [59]. Comparative analysis by Kause, Smith, Prytherch, and Parr (2004) highlighted a disparity in unplanned Pediatric Intensive Care Unit (PICU) admissions, with the United Kingdom reporting a rate of 47.3% as opposed to New Zealand’s 24.2% [6]. These unplanned admissions are associated with several adverse outcomes, including extended duration of mechanical ventilation [15], prolonged stays in the PICU, escalated mortality rates [17,18,19,60], increased severity of illness, and augmented healthcare costs [16]. Unplanned transfers, according to Christopher P. Bonafide and colleagues (2014), are associated with a 4.97-fold increased chance of death (95% CI, 3.33–7.40; *p* = 0.001) [61]. It is a low-quality metric [3]. The literature indicates that the adoption of pediatric rapid response teams reduced unplanned cardiac ICU transfers from 16.8 to 7.1 per 1000 patients per day, with a p value of 0.015 [42]. Panesar et al., (2014) reported identical findings as well [62]. The duration of stay in the PICU was reduced (19.0%), and mortality was also reduced (22%). After rapid response team implementation, the relative risk of death following unplanned admission to the PICU was 0.685 [63]. Notably, 59.66% of our rapid response calls resulted in PICU admissions in our research, which is higher than the published data range of 30% to 57% [16,64,65,66] and a similar 59.6% [67]. However, prior to the rapid response team initiative, 63% of unplanned PICU admissions from the ward occurred, whereas this decreased to 33.57% when a modified PWES score-based pediatric rapid response team intervention was implemented. Furthermore, before and after intervention, PICU admissions from the general pediatric ward were 30.5% (487/1604) versus 16.8% (281/1677). Hence, unadjusted relative risk of unplan PICU admission is 0.55 (0.48–0.63), *p* value 0.0001 (Table 3). We were able to cut unexpected PICU admissions by 45%, a statistically meaningful reduction. During our investigation, interrupted time series (ITS) analysis was employed to evaluate the efficacy of the pediatric rapid response team (RRT) intervention, as depicted in Figure 2. This intervention was demonstrably associated with a reduction in the rate of unplanned admissions to the Pediatric Intensive Care Unit (PICU). Specifically, the analysis yielded a coefficient of −9.61, with a 95% confidence interval ranging from −12.65 to −6.57 and a statistically significant *p*-value of 0.001, as detailed in Table 5. This finding suggests that the implementation of the RRT intervention led to a significant decrease in unplanned PICU admissions, averting approximately 9.61 such admissions per unit of time analyzed.

Cardiac arrest outside of the ICU is a severe, clinically significant, and frequently fatal event [68,69,70]. An event like this is frequently preceded by indications that a person’s health is deteriorating. This implies that it is not always a sudden or unexpected occurrence [71,72]. When a patient exhibits clinical instability, prompt intervention may reduce the number of cardiac arrests and fatalities [35]. According to Berens (2011), 2–6% of pediatric patients suffer an in-hospital cardiac arrest. Children who have a cardiopulmonary arrest in the hospital have a poor survival rate (25–38%) [73]. Our analysis further clarified that the implementation of the pediatric rapid response team (RRT) intervention was significantly associated with a reduction in the incidence of non-ICU cardiac arrests. This relationship was quantified by a coefficient of −1.641, indicative of 1.64 fewer events per unit of time, within a 95% confidence interval ranging from −2.22 to −1.06 and a *p*-value of 0.001, as presented in Table 5. These findings robustly support the effectiveness of the RRT intervention in improving critical pediatric healthcare metrics. The positive impact of the RRT intervention on pediatric healthcare outcomes has been affirmed by several studies. Brilli RJ, Gibson R, and colleagues (2007) reported a relative risk (RR) of 0.41, with a 95% confidence interval from 0 to 0.86 and a p-value of 0.02 [74]. Similarly, other researchers observed a RR of 0.62 (95% CI: 0.45 to 0.84) [75], and noted findings regarding cardiorespiratory arrest yielded a RR of 0.64, with a 95% confidence interval of 0.55 to 0.74 among the pediatric population. Furthermore, Maharaj R (2015) [76] and Trubey R and associates (2019) documented a RR of 0.27, with a confidence interval of 0.07 to 0.95 [77]. Paul J. Sharek and his team also demonstrated a significant decrease in non-ICU cardiorespiratory arrests post-intervention, with the rate per 1000 admissions in the post-intervention cohort being 0.29 times that of the pre-intervention group (95% CI: 0.10 to 0.65; *p* = 0.008), highlighting the significant role of the intervention in enhancing pediatric healthcare outcomes [40].

Moreover, Chong SL and Goh MSL (2022) discovered identical results [78]. Additionally, Saad Al-Qahtani (2013) proclaimed that non-ICU cardiopulmonary arrests decreased from 1.4 to 0.9 per 1000 hospital admissions (relative risk, 0.68; 95% confidence interval, 0.53–0.86; *p* = 0.001) [79]. Joffe et al. discovered no difference (2011) [80], and Tibballs J (2009) reported that unexpected cardiac arrests did not change from 0.19 per 1000 to 0.17 per 1000 (risk ratio 0.91, 95% CI (0.50–1.64, *p* value 0.75) [65]. In conclusion, a systematic review and meta-analysis on this topic demonstrated that the incidence of cardiorespiratory arrests outside of the ICU was significantly reduced, with a relative risk (RR) of 0.62 and a 95% confidence interval (CI) ranging from 0.46 to 0.84 and the *p*-value was 0.35 [81].

Interestingly, we found that out of 1674 patients treated following the intervention, not even one had a code blue, but there were 24 cardiorespiratory arrests before the program was introduced. Consequently, the pediatric rapid response team intervention dropped the cardiac arrest from 24 to 0 incidents when compared to non-ICU cardiopulmonary arrests before and after the intervention. The intervention was successful in enhancing patient safety because there was no relative risk (RR = 0) (Table 4). In our study, the mortality contrast between the pre- and post-exposure periods is striking. Of the 1604 infants admitted during the pre-exposure epoch, 36 succumbed post-transfer to the PICU. This is juxtaposed against the subsequent phase, where merely 3 out of 1677 patients met a similar fate. This translates to a mortality rate of 22.5 per 1000 admissions in both the periods antecedent and subsequent to the initiative’s introduction. The derived unadjusted risk ratio stands at 0.08, underpinned by a confidence interval of (0.02–0.25) and a highly significant *p*-value of 0.0001. Our findings resonate with the broader academic discourse. Sharek PJ (2007) observed a congruent trend, wherein the mean monthly mortality rate registered an 18% descent, moving from 1.01 to 0.83 deaths per 100 discharges, demarcated by a 95% CI of (5–30%) and a *p*-value of 0.007 [36]. E. J. Sandhu et al., (2015) presented mortality figures reflecting a RR of 0.79, enclosed within a 95% confidence interval of 0.63–0.98 [81]. Maharaj R. (2015) echoed these findings, tabulating a mortality RR of 0.82, with a 95% CI bracketed between 0.76–0.89 [76]. In a later study, Santhanam S. (2018) documented an overarching hospital mortality rate with an RR of 0.79, emphasizing a CI spanning 0.63 to 0.98 [75]. Gong XY (2020) further enriched the literature with evidence accentuating a significant dip in hospital mortality (*p* = 0.025) [82]. Bolstering these narratives, a meticulous systematic review and meta-analysis inferred that the institution of a pediatric rapid response team is invariably tethered to a substantial decrement in hospital mortality, characterized by an RR of 0.88 and a 95% CI of (0.83–0.93) [83]. Notably, a departure from this prevailing consensus was registered only by Joffe AR and Anton NR (2011), who reported a neutral impact on hospital mortality [80]. Drawing from these rigorous empirical engagements and our findings, the critical role of pediatric rapid response team interventions in mitigating pediatric mortality emerges unequivocally.

The sensitivity analysis highlights the diversity in the impact of the pediatric rapid response team’s (RRTs) intervention on unscheduled PICU admissions. Despite a mean admission decrease of −9.61 in the initial effect, there is significant variation suggested by the broad 95% confidence interval (−3.65 to 4.04), suggesting that the RRTs effect is inconsistent and could be impacted by factors like data noise and sample size (Figure 3) This diversity is corroborated by the segmented regression analysis, which further emphasizes how difficult it is to measure the intervention’s efficacy in decreasing unplanned PICU admissions. On the other side, there is a clear positive effect when looking at non-ICU cardiac arrest data using the zero-inflated Poisson (ZIP) model. The intervention’s ability to minimize non-ICU cardiac arrests is shown by the significant increase in the likelihood of zero arrests post-intervention (estimate = 5.081, *p* 0.001). These results show how difficult it is to evaluate healthcare treatments. Further research with larger datasets or proper research technique is needed to clarify the influence of the pediatric RRTs on unplanned PICU admissions, as it shows promise in lowering non-ICU cardiac arrests.

Clinical implication: Our research is unique because of the novel methodology we used to validate our findings; we used ITS and then a sensitivity test. Through this method, in conjunction with the detailed depiction of the several charts, our study provides valuable insights into intervention efficacies and sets a high-level precedent for future efforts in evidence-based pediatric care research through the integration of these advanced statistical tools. Additionally, Rapid Response Teams (RRTs) greatly enhance patient outcomes in pediatric care, according to the study’s conclusions. To put this into practice, it may be necessary to incorporate RRTs into routine pediatric care protocols to lessen the number of children admitted to the PICU without a valid reason and to forestall cardiac arrests, often known as Code Blue episodes. It is important for hospitals to think about staff training, creating clear criteria for RRT activation, and making sure they could respond quickly. Health outcomes, critical event rates, and patient safety in pediatric wards might all see an improvement with an emphasis on RRTs in patient care. It stresses the importance of ongoing monitoring and quality improvement, and it addresses the need for specialized training to activate RRTs according to defined protocols. Not only do the dramatic drops in cardio-pulmonary arrests and PICU hospitalizations establish that RRTs work, but they also lay the groundwork for healthcare policy and budget allocation, encouraging a culture of constant improvement and patient safety.

Study Limitations: This study’s brief duration is one of its limitations. A thorough grasp of the long-term effects of the pediatric rapid response team on unanticipated PICU admissions and non-ICU cardiorespiratory arrests may not be possible given the short duration of the research. The fact that the study is based on observational studies, which are prone to bias and confounding and might not accurately reflect the true impact of the pediatric rapid response team, is a second limitation of the research. Thirdly, the generalizability of the study results outside the hospital where it was performed was constrained using single-center data. The results could be influenced by the pediatric patients’ demographic, organizational, and clinical traits. Fourth, the impact of the pediatric rapid response team for reduction of hospital-wide mortality may not be accurately measured by the relative risk that was calculated without accounting for confounding factors. There are many significant missing variables in the research, such as the pediatric rapid response team’s median response time, the patient outcomes following the activation of the rapid response team, the number of patients who remain in the ward, and their outcomes. The inability to compute the PRISM score for transferring patients from the ward restricted the ability to manage the impact of confounding variables. Another limitation of the research is lacking data on the compliance rate of the pediatric rapid response team activation, based on modified early warning signs. It might have affected the study’s results and prevented them from being extrapolated to other contexts where compliance rates might vary. Finally, the sensitivity test did not support the validity of our study finding, particularly the impact on the reduction of unplanned PICU admission. Our research is unique because of the novel methodology we used to validate our findings; we used ITS and then a sensitivity test. Through this method, in conjunction with the detailed depiction of the several charts, our study provides valuable insights into intervention efficacies and sets a high-level precedent for future efforts in evidence-based pediatric care research through the integration of these advanced statistical tools.

## 5. Conclusions

The conclusive evidence from this study indicates that the implementation of pediatric rapid response teams (RRTs) is not merely a correlational factor but a driving force in enhancing pediatric healthcare outcomes. The significant reductions in unplanned PICU admissions and non-ICU cardiopulmonary arrests attest to the efficacy of RRTs. While further research is essential to refine these findings, the compelling data underscore the potential of RRTs in transforming pediatric care, advocating for their broader integration and rigorous analysis within clinical settings to solidify their role in patient care improvement.

## Figures and Tables

**Figure 1 healthcare-12-00518-f001:**
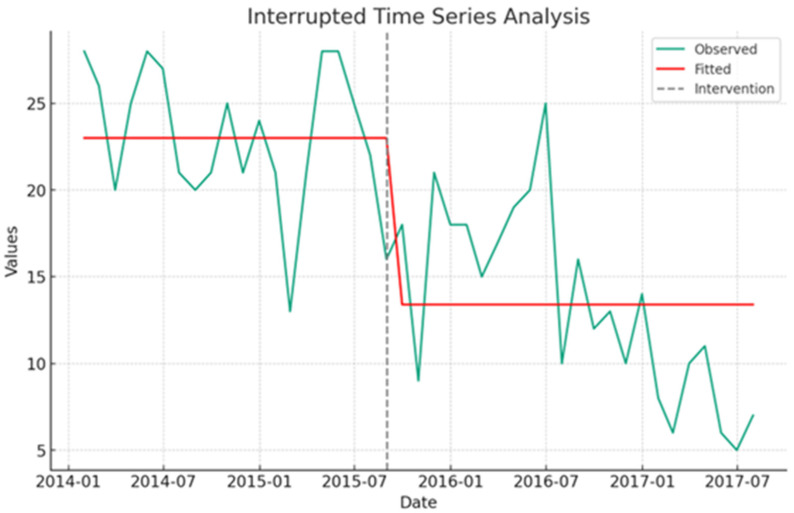
This figure illustrates the results of the Interrupted Time Series (ITS) analysis focusing on the impact of the pediatric rapid response team intervention on unplanned Pediatric Intensive Care Unit (PICU) admissions. The plot displays the observed un-planned PICU admission rates over time alongside the fitted values predicted by the ITS model. A vertical grey dashed line indicates the intervention’s implementation in September 2015, demarcating the pre- and post-intervention phases. The trend lines provide a visual representation of unplanned PICU admissions, demonstrating a notable decrease following the introduction of the RRTs intervention.

**Figure 2 healthcare-12-00518-f002:**
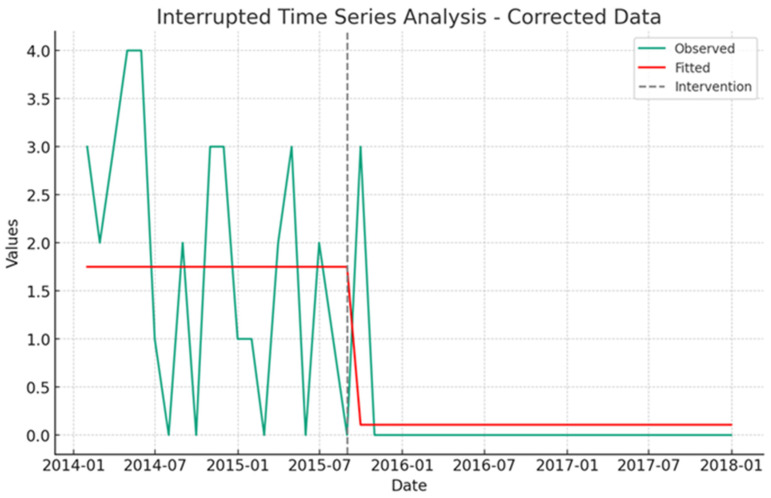
An Interrupted Time Series (ITS) analysis displays the monthly counts of non-ICU cardiorespiratory arrests in pediatric patients from January 2014 to July 2017, with observed data and fitted model values illustrated in distinct line styles. The implementation of the pediatric rapid response team in September 2015 is denoted by a grey dashed line, serving as a demarcation for comparing trends and assessing the intervention’s impact in the pre- and post-intervention periods.

**Figure 3 healthcare-12-00518-f003:**
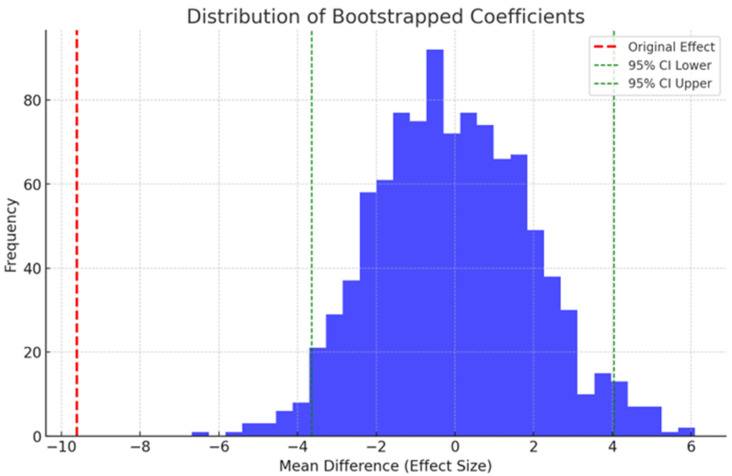
Sensitivity test by bootstrapped coefficient distribution for unplanned PICU admissions. This histogram depicts the distribution from 1000 bootstrapped samples analyzing the impact of a healthcare intervention on unplanned PICU admissions. The original mean difference of −9.61 (red dashed line) suggests a reduction in admissions post-intervention. The 95% confidence interval, ranging from −3.65 to 4.04 (green dashed lines), indicates substantial variability in the intervention’s effect.

**Table 1 healthcare-12-00518-t001:** King Abdulaziz University Hospital, Heddah, Saudi Arabia, Pediatric Early Warning System escalation of care algorithm as per colour code. pRRT = pediatric rapid response team.

Score = 0–3	Score = 4–6	Score ≥ 7
Record observational findings at least once every 4 h. Inform Primary Medical team. Manage fever, pain, fluids, or distress. Review O_2_ requirements.	Record observational findings at least once every 1 h. Inform Charge Nurse. Inform Primary Medical team. Manage fever, pain, fluids, or distress. Review O_2_ requirements. Consider pRRT activation.	Record observational findings at least once every 30 min. Inform Charge Nurse. Inform Primary Medical team/most responsible physician review. Manage fever, pain, fluids, or distress. Review O_2_ requirements. Immediate activation of pRRT.

**Table 2 healthcare-12-00518-t002:** Comparative outcomes pre- and post-intervention. Both cohorts exhibited a consistent PEWS median score of 6 (*p* = 0.217). Demographic variables, including median age and gender, showed no significant disparities (*p* = 0.31 and *p* = 0.45, respectively). However, metrics like ward to PICU admission (WdToPICUAd), inPtcode events (pediatric code outside of ICU), PICU mortality, and MortalityWAd (mortality of PICU patients with unplanned transfer from ward) were notably significant (*p* < 0.05 for each).

Variable	Pre-Intervention	Post-Intervention	*p*-Value
Date range	1 January 2014 to 30 August 2015	1 September 2015 to 30 July 2017	x
Total number of months	22	21	x
PEWS score (median [IQR])	6.00 [5.00, 6.75]	6.00 [5.00, 7.00]	0.217
Age, median (IQR)	6 (IQR 2–12)	7 (IQR 3–24)	0.31
Gender, Female (%)	725 (45.2%)	780 (46.5%)	0.45
Ward admission (median [IQR])	72.50 [67.25, 79.75]	80.00 [78.00, 82.00]	0.009
WdToPICUAd (median [IQR])	21.50 [20.25, 25.75]	13.00 [10.00, 18.00]	<0.001
inPtcode (median [IQR])	2.00 [0.25, 3.00]	0.00 [0.00, 0.00]	<0.001
PICU Mortality (median [IQR])	3.00 [2.00, 3.00]	1.00 [1.00, 2.00]	0.001
MortalityWAd (median [IQR])	2.00 [0.25, 2.00]	0.00 [0.00, 0.00]	<0.001

**Table 3 healthcare-12-00518-t003:** A two-by-two contingency table evaluating the impact of a pediatric rapid response team on unplanned PICU admissions. Comparing pre- and post-intervention periods, the data reveals a notable reduction in admissions, with an estimated relative risk of 0.552 and a 95% confidence interval of 0.485 to 0.628, demonstrating a significant decrease (*p*-value < 0.0001). This highlights the RRT’s effectiveness in enhancing patient safety and quality of care in high-impact healthcare settings.

	PICU Admission	Non PICU Admission	Total
Exposed (Post Intervention)	281	1396	1677
Un-Exposed (Pre-Intervention)	487	1117	1604
Total	768	2513	3281
Results: Estimate 0.552, 95% CI (0.485–0.628), *p*-value 0.0001			

**Table 4 healthcare-12-00518-t004:** This table compares non-ICU cardiopulmonary arrests before and after the pediatric rapid response team intervention, showing a reduction from 24 to 0 cases. This complete elimination post-intervention (RR = 0) highlights the intervention’s effectiveness in im-proving patient safety.

College	Cardiopulmonary Arrest	No Cardiopulmonary Arrest	Total
Pre-Intervention	24	1580	1604
Post-Intervention	0	1674	1674
Total	24	3254	3278
Results: Since there are 0 events in the post-intervention group, the RR would be 0. This indicates a complete reduction in the event rate (non-ICU cardiopulmonary arrest) post-intervention.			

**Table 5 healthcare-12-00518-t005:** Interrupted Time Series analysis results, evaluating pediatric rapid response team effects on unplanned PICU admissions and non-ICU cardiopulmonary arrests. Coefficients indicate change in outcomes post-intervention; negative values denote reductions. Standard errors assess estimate precision, *t*-values test the effect significance, and 95% confidence intervals indicate probable coefficient range. *p*-values (<0.001) demonstrate strong evidence against the null hypothesis. Results suggest significant reductions in both unplanned PICU admissions and non-ICU cardiopulmonary arrests post-RRTs implementation.

Variable	Coefficient	Std Error	*t*-Value	95% CI	*p*-Value
Unplanned PICU Admission					
Intercept	23.00	1.100	20.97	20.78–25.22	<0.001
RRTs action	−9.61	1.504	−6.391	−12.65 to −6.57	<0.001
Non-ICU Cardiopulmonary Arrest (Code Blue)					
Intercept	1.750	0.220	7.963	1.31–2.19	<0.001
RRTs action	−1.641	0.288	−5.71	−2.22 to −1.06	<−0.001

## Data Availability

Data will be supplied on valid reason on request by the author.

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
