# Peer review of "Impact of Rapid Response Teams on Pediatric Care: An Interrupted Time Series Analysis of Unplanned PICU Admissions and Cardiac Arrests"

_healthcare, 2024, doi:10.3390/healthcare12050518_

Round 1
Reviewer 1 Report
Comments and Suggestions for Authors
Here are some comments and suggestions:
- In abstract it is necessary to provide information about the variables and the statistical analysis carried out.
- Keywords should be MeSH terms.
- Regarding this statement in the introduction "According to Children's Health Coalition of America (CHCA) statistics, between 2003 and 2006" Is this the latest data?
- Improve the clarity of the objective at the end of the introduction.
- ​The design refers to a quasi-experimental cohort. Be careful because it can be confused with proposing a cohort study, which would not be the case. Therefore, try to clarify it.
- In ethical aspects, reference should be made to the Helsinki declaration and the code of good research practices. In addition, mention must be made of the confidentiality of the data collected and the consent of the parents or legal guardians.
- Provide sensitivity and specificity data in the validation of the scales used in the methodology in the subsection "Activation and Scoring Mechanism"
- Clearly indicate the dependent and independent variables and describe them.
- The results provide OR data, among others, but this analysis has not been indicated in the statistical analysis. It is also necessary to provide information on the contrast tests performed according to the normality of the sample.
- ​The methodology must clearly explain the intervention
- With the sample provided, why are the interquartile ranges provided and not the mean and standard deviation? Explain in methodology the use of parametric tests or not according to normality and the test for the determination. Review.
- Tables must have a title
- Try to improve the writing and clarity of the discussion. Above all to avoid such dense paragraphs using more periods and paragraphs.
- The conclusion is not clear. The proposed objective must be responded positively or non-positively.
Comments on the Quality of English Language
Author Response
Dear Sir/Madam,
Thanks for review my article.
I have edited my manuscript as your recommendations.

Reviewer 2 Report
Comments and Suggestions for Authors
First of all, I would like to express my gratitude for the opportunity I have been given to review Samah Al Harbi's manuscript, which aims to evaluate RRTs' effectiveness in decreasing cardiac arrests and unexpected Pediatric Intensive Care Unit (PICU) admissions.
Comments and suggestions:
-The title of the work is too long and does not adequately summarize the content of the article
-In the abstract, the headings of each of the parts into which it is divided must be eliminated, in accordance with the journal's standards.
-In the backgroundd section of the abstract, authors discuss what is expected of RRTs , but nothing is commented on what is known about the topic.
-In the methodology section of the abstract, authors should provide more information on how the work has been carried out.
-The introduction is too short and generic, not adequately contextualizing the topic studied. Nothing is said about RRTs or what has been published so far in this regard. It would be necessary for the authors to provide new studies that support and make the introduction more solid.
The methodology is unclear, so the authors must redo it taking into account the following suggestions:
-The authors should better explain the reason why they consider it to be a quasi-experimental study.
-The authors must specify whether or not all pediatric patients admitted to the unit are included, regardless of the cause.
-The authors must explain whether the ethical recommendations for the development of quasi-experimental research work have been met.
-It is not at all clear what intervention has been carried out.
-Is there more than one RRTs in the hospital? Has the possible variability in the assessment between the different RRTs, if any, been taken into account? How has it been corrected?
-The authors must provide more information regarding how, where, and who carried out the data collection procedure.
-The main and secondary results of the study should be stated in greater detail
-The authors should better explain how the results found in this study can influence clinical practice. They should also comment on whether there are practical implications or specific recommendations derived from the results.
-The conclusion must be expanded taking into account the results obtained
Comments on the Quality of English LanguageModerate editing of English language required
Author Response
Dear Sir/Madam,
Thanks for review my article.
I have edited my manuscript as your recomendations.

Reviewer 3 Report
Comments and Suggestions for Authors
Dear author,
The description and presentation of methods and results is not clear because the quality of the English used needs editing; there are grammatical errors, typos and sentence constructions that are not the most appropriate. You must, therefore, review the entire text. Here are just some examples:- 51. … morality (mortality)
- 79. … comma and no period before “as well as”
- 103. … dispensed with.
- 108. …. Augmented (supplemented by?)
- 147. … tableone (table one)
- 186. … admission 0.55 …. (admission was 0.55??)
- 330. … review grammar
- 331. … review grammar
- 349/50. … review phrase grammar
- 368/70. … review phrase grammar
- 379/81. … review phrase grammar
- 470. … with ??
Discussion is very long. There is no need to repeat the data mentioned in the results again; The important thing is to explain the reasons for the differences found in previous studies.
Review the first sentence of the conclusion as it is not clear. The conclusion deserves further refinement.
The study period took place more than 7 years ago (pre-intervention: January 1, 2014, to August 30, 2015, and post-intervention: September1, 2015, to July 30, 2017). Do you know if the current data is in line with those found in the post-invention phase at the time of the study?
Although approval was obtained from the local ethics committee, as the data were collected prospectively during the intervention period, I believe that informed consent should have been applied, regardless of the retrospective analysis that was subsequently carried out. In future studies, you should consider obtaining informed consent from patients/legal representatives undergoing follow-up at least when collecting data prospectively.
Regards
Comments on the Quality of English LanguageSee above, please
Author Response
Dear Reviewer,
Thank you for your constructive feedback on our manuscript. I have carefully reviewed your comments and have undertaken a thorough revision of the text to address the concerns regarding the clarity, grammar, typographical errors, and sentence construction. Below, we summarize the revisions made in response to each of the examples you provided, as well as additional edits to enhance the overall quality of our manuscript.

Round 2
Reviewer 1 Report
Comments and Suggestions for Authors
The aspects discussed have been clarified